# Is Incisor Compensation Related to Skeletal Discrepancies in Skeletal Class III? A Retrospective Cephalometric Study

**DOI:** 10.3390/diagnostics14101021

**Published:** 2024-05-15

**Authors:** Jirath Mathapun, Chairat Charoemratrote

**Affiliations:** Department of Preventive Dentistry, Faculty of Dentistry, Prince of Songkla University, Hat Yai 90110, Thailand; saverry@hotmail.com

**Keywords:** skeletal Class III, jaw position, incisor compensation, incisor inclination, lateral cephalogram

## Abstract

This study investigated compensation in skeletal Class III subjects to compare various severities of abnormal jaws. A retrospective analysis of 137 skeletal Class III cephalograms (63 males and 74 females) was conducted, with cephalometric assessments determining skeletal and dental values. The results were compared with Class I cephalograms. Incisor compensation was examined by pairing normal jaws with varied abnormal jaws, classified by severity using one standard deviation (SD). Statistical analyses included Kruskal–Wallis tests, Bonferroni tests, Spearman’s correlations, and multiple linear regression. Four skeletal Class III groups were identified: OMx+PMd, RMx+OMd, OMx+OMd, and PMx+PMd (P = prognathic; O = orthognathic; R = retrognathic; Mx = maxilla; Md = mandible.). The upper central incisor (U1) showed proclination, and the lower central incisor (L1) showed retroclination across all groups except for U1 in PMx+PMd and L1 in OMx+OMd, which exhibited normal inclination. U1 exhibited limited compensation even with progressive maxillary retrognathism, while L1 showed limited compensation after one SD of mandibular prognathism. Maxilla (SNA) and jaw discrepancy (ANB) were inversely related to the U1 degree, whereas only jaw discrepancy (ANB) was positively related to the L1 degree. U1 in PMx+PMd and L1 in OMx+OMd showed no incisor compensation. U1 had limited compensation even with progressive maxillary retrognathism while L1 showed limited compensation after one SD mandibular prognathism.

## 1. Introduction

Skeletal Class III is characterized by maxillary and mandibular anteroposterior discrepancies that present as retrognathic maxilla, prognathic mandible, or a combination of both [1,2,3,4,5,6,7,8]. The occurrence of skeletal Class III malocclusion exhibits significant variation across countries, regions, and ethnicities, with prevalence rates that range widely [9,10,11,12,13,14,15,16]. The lowest rate of 2.5% was reported in Australia [9], while the highest rate of 31.4% was reported in South Korea [16]. When this anteroposterior malrelation is presented, the incisors try to move toward each other with respect to a function known as incisor compensation [17]. Generally, the upper incisors procline toward the lip [18,19,20], whereas the lower incisors lean toward the tongue [18,19,20,21] in skeletal Class III patients. In a severe prognathic mandible, the lower incisors are regularly found to be retroclined [22,23]. Interestingly, the lower incisors could be in normal inclination relative to the moderately retrusive maxilla with a slightly protrusive mandible [23]. This could imply that a more prognathic mandible would result in the greater retroclination of the lower incisors. Nevertheless, the association between the degree of a prognathic mandible and the inclination of the lower incisors has not been investigated. Unfortunately, the association between the degree of a prognathic mandible and the inclination of the lower incisors has not been investigated. In the case of a retrognathic maxilla, the upper incisors exhibited proclination. Again, the normal inclination of the upper incisors was reported in an orthognathic maxilla with a prognathic mandible and in a severely retrusive maxilla with an orthognathic mandible [23]. However, the association between the degree of retrognathic maxilla and the inclination of the upper incisors has not been studied.

The amount of anteroposterior discrepancy in skeletal Class III is the focus of this research in order to determine its potential impact on the degree of compensation. Furthermore, incisor compensation in different abnormal jaw positions with opposite normal jaws has never been explored.

Previous skeletal Class III studies have focused on specific clusters based on the maxilla and mandibular positions, but not all have universally observed incisor inclination [22,23,24,25,26]. Moreover, previous skeletal Class III studies were conducted according to the maxilla and mandibular anteroposterior positions in either an orthognathic [23,24] or retrognathic maxilla [22,23,24] with either orthognathic [23] or prognathic mandibles [22,23]. However, skeletal Class III cases with a prognathic maxilla and mandible have never been researched and are only presented as case reports [27,28,29]. The three gaps in knowledge can be summarized as follows: (1) Incisor compensation in skeletal Class III with different maxilla and mandibular positions is still controversial; (2) the association between the degree of skeletal discrepancy and incisor compensation has not been studied; (3) skeletal Class III discrepancies should be grouped according to maxilla and mandibular positions, and incisor compensation should be studied. The aims of this study were to group skeletal Class III anteroposterior discrepancies according to the maxilla and mandibular positions and study their incisor inclinations. Furthermore, the association between the degree of skeletal discrepancy and incisor inclination was investigated based on pairing one normal jaw with different degrees of severity relative to the opposite abnormal jaw. Finally, the study examined the relationships between incisor inclination and the investigated parameters.

## 2. Materials and Methods

### 2.1. Subjects

This retrospective study was conducted following the approval obtained from the Institutional Review Board for human patients (protocol EC6410-064) at the Faculty of Dentistry, Prince of Songkla University. All cephalograms diagnosed as skeletal Class III were meticulously selected from a pool of patients treated between 2011 and 2021. The inclusion criteria comprised non-growing patients aged 18–35 years with good-quality lateral cephalometric radiographs. Patients with a history of (1) previous orthodontic treatment or orthognathic surgery, (2) prostheses or extensive restorations on the anterior teeth, (3) periodontal and gingival diseases, (4) facial trauma or plastic surgery in the facial region, and (5) systemic diseases or endocrinopathies were excluded from the study to ensure a homogenous sample and minimize confounding variables.

### 2.2. Study Sample Calculation and Group Designations

The sample size calculation, which relied on the mean and standard deviations extracted from an article by Li et al. [22], utilized G*Power software version 3.1 with an alpha of 5% and a power of 80%. The calculation recommended a minimum of 11 samples per group to achieve reliable results. Consequently, all cephalograms diagnosed as skeletal Class III from 2011 to 2021 were scrutinized, which resulted in the classification of patients into four distinct groups based on jaw discrepancies, with each group containing at least 11 films. These four groups were designated as follows: orthognathic maxilla and prognathic mandible (OMx+PMd), retrognathic maxilla and orthognathic mandible (RMx+OMd), orthognathic maxilla and mandible (OMx+OMd), and prognathic maxilla and mandible (PMx+PMd). Additionally, 30 subjects exhibiting skeletal Class I and normal occlusion were included as the norm for comparative purposes.

The study employed a design wherein one normal jaw was paired with varying severity levels of the opposite jaw to facilitate a comprehensive comparison of incisor compensation. The severity of the opposite jaw was meticulously assessed, and 1 standard deviation (SD) was utilized to categorize the severity levels. A comparative study between OMx+OMd and OMx+PMd was introduced, with PMd being further stratified into OMx+PMd (1 SD) and OMx+PMd (2 SDs). Additionally, a comparative study between OMx+OMd and RMx+OMd was proposed; however, the RMx presented only 1 SD without 2 SDs; therefore, the study was not replicated as these cases were already included in the comparison among the four groups. This approach ensured the comprehensive exploration of incisor compensation patterns and provided valuable insights into the complex dynamics of skeletal Class III malocclusions.

### 2.3. Lateral Cephalometric Assessments

In lateral cephalometric assessments, strict adherence to standardized procedures ensured the accuracy and reliability of the data collected. The natural head position served as a consistent reference point for all cephalograms that provided a reliable framework for analysis and comparison. The Orthopantomograph OP300 (Instrumentarium Dental, Tuusula, Finland) was employed with settings of 90 kV, 12.5 mA, and a 15 s exposure time. Uniform imaging conditions were maintained across all captured images that minimized potential variations in image quality and interpretation.

Subsequently, each cephalogram underwent meticulous digitization and analysis using Dolphin Imaging^®^ (version 11.9; Dolphin Imaging, Chatsworth, CA, USA). To ensure precision in measurements, cephalogram data were converted into actual distances. This conversion process involved utilizing the scale ruler embedded within the cephalogram with mathematical calculations performed using the ImageJ software, version 1.53a (NIH, Bethesda, MD, USA).

The comprehensive linear and angular measurements assessed on lateral cephalograms are presented in detail in Table 1, providing an overview of the parameters analyzed in the study.

### 2.4. Statistical Analysis

A single examiner blinded to the corresponding film of the same subject executed all measurements, ensuring an unbiased and consistent approach. Prior to the measurements, the researcher compared the measurements with an expert to determine the interclass correlation. To assess measurement error and reliability, a subset of 30 randomly chosen subjects underwent remeasurement after a two-week interval. The evaluation of reliability involved independent t-tests and interclass and intraclass correlation coefficients. Notably, systematic errors were absent in the paired *t*-test (*p* < 0.05), which affirmed the precision and consistency of the examiner’s assessments. Furthermore, random errors were estimated using the Dahlberg formula.

The Shapiro–Wilk test unveiled the absence of normally distributed variables within the dataset. Consequently, to examine the mean differences between the four groups and skeletal Class I, the analysis proceeded with the Kruskal–Wallis test followed by the Bonferroni test. This non-parametric approach was selected due to the non-normal distribution of variables.

The relationships between U1-NA and the other measurements were examined within the RMx+OMd combined with OMx+OMd groups. Similarly, the relationships between L1-NB and the other measurements were scrutinized within the OMx+PMd combined with OMx+OMd groups. Spearman’s correlations and multiple linear regression analyses were employed to discern the variables influencing U1-NA (deg) and L1-NB (deg), thereby shedding light on the intricate associations between these parameters and other measurements. The statistical analyses were conducted using SPSS version 17 (SPSS, Chicago, IL, USA). The level of significance for all tests was set at *p* < 0.05.

## 3. Results

The interclass correlation coefficient was 0.92, and the intraclass correlation coefficient was registered at 0.95, which underscored the excellent reliability of measurements. Furthermore, random errors estimated using the Dahlberg formula indicated measurement errors of 0.87 mm for linear measurements and 0.62 degrees for angular measurements. These values were well within the acceptable ranges and affirmed the reliability and accuracy of the measurement methodology.

The prevalence of skeletal discrepancy among skeletal Class III subjects revealed its highest occurrence in the OMx+OMd group, which accounted for 45.26% of the cases (Figure 1 and Table 2). In descending order, prevalence was observed in the OMx+PMd, RMx+OMd, and PMx+PMd groups. The examination of the SNA and SNB values indicated jaw malpositions were categorized into subgroups. Notably, the most substantial jaw discrepancy was evident in the OMx+PMd subgroup, while the PMx+PMd subgroup exhibited the least disparity. Moreover, the assessment of the vertical relationship unveiled distinct characteristics within the subgroups. Specifically, the OMx+PMd and RMx+OMd subgroups demonstrated normodivergent profiles, while both the OMx+OMd and PMx+PMd subgroups exhibited hyperdivergent patterns.

In comparison with the skeletal Class I group, it was evident that most skeletal Class III groups revealed greater U1 proclination with the exception of the PMx+PMd group, which exhibited standard inclination. Conversely, when assessing U1-PP, all groups demonstrated a regular inclination compared to the skeletal Class I group. Moreover, a significant finding emerged as most groups displayed considerable L1 retroclination in contrast to the skeletal Class I group with the most pronounced retroclination observed in the OMx+PMd subgroup. However, the OMx+OMd subgroup maintained a standard L1 inclination. Across all skeletal Class III groups, a normal L1 position was consistently observed (Table 3 and Table 4).

In this study, three distinct groups characterized by varying degrees of mandible prognathism were systematically occluded with OMx (Table 5 and Table 6). Across all groups, a consistent observation emerged. The U1 exhibited uniform proclination and protrusion, which indicated a stable pattern of dental positioning irrespective of mandibular prognathism severity. In the OMx+OMd group, the L1 exhibited the least retroclination in comparison to OMx+PMd at one SD and two SDs where more pronounced retroclination was evident. Surprisingly, both PMd groups displayed no significant difference in L1 inclination.

An estimate equation to determine U1 inclination, considering U1-NA (mm), U1-PP, ANB, and SNA, was formulated using multiple linear regression analysis coefficients:U1-NA (deg) = −23.85 + 0.715 (U1-NA (mm)) + 0.556 (U1-PP) − 0.365 (ANB) − 0.237 (SNA)

This equation indicates that U1-NA (degree) will have increased inclination with higher degrees of U1-NA (mm) and U1-PP. Conversely, U1-NA (degree) will have decreased inclination as the degrees of ANB and SNA increase.

Similarly, the inclination of L1-NB (degree) can be determined using multiple linear regression analysis coefficients:L1-NB (deg) = −18.364 + 0.467 (L1-MP) + 0.539 (L1-NB (mm)) + 0.532 (ANB)

This equation suggests that L1-NB (degree) will have increased inclination with higher degrees of L1-MP, L1-NB (mm), and ANB.

## 4. Discussion

Skeletal Class III in this study was categorized based on jaw discrepancy with a minimum number of 11 films per group. Four groups that met these criteria were identified: PMx+PMd, OMx+PMd, OMx+OMd, and RMx+OMd. An additional group of RMx+RMd (*n* = 4) was presented but was excluded from the analysis. The compensation patterns of U1 (proclination) and L1 (retroclination) observed in OMx+PMd and RMx+OMd were the same as previously reported [22,23]. This study introduced two additional groups: OMx+OMd and PMx+PMd. The OMx+OMd group exhibited U1 proclination with L1 maintaining a normal inclination, whereas PMx+PMd showed normal U1 inclination with L1 retroclination. Unfortunately, no prior research is available for either OMx+OMd or PMx+PMd, which makes direct comparisons challenging. This study revealed that normal inclination could be found in specific skeletal Class III conditions.

In this study, one normal jaw was paired with different severities of the opposite jaw to compare incisor compensation. The orthognathic mandible with RMx showed U1 proclination and protrusion even though the mandibles were progressively anteriorly positioned, which indicated that U1 has limited compensation. The cause of this limited movement could be the equilibrium position of the tooth from the function of the upper lip [30].

When we paired an orthognathic maxilla with PMd in a contrasting investigation where the mandible protruded by one SD, a significant retroclination of 7.85 degrees was observed. However, after two SDs, the mandibular position increased, and compensation ceased. The initial compensation could be explained according to a previous study [17] that reported that the compensation mechanism existed to allow the upper and lower incisors to be accommodated in a normal relationship. Interestingly, no more retroclination occurred at two SDs, which could be considered as limited compensation; this may result from the mandibular incisors that are located within a narrow zone for the purposes of establishing equilibrium between the tongue and lips [31]. A different explanation could have to do with the force’s direction. Since the mandibular incisors are positioned somewhat parallel to the jaw’s closing arc, less movement is produced by the occlusal force, which can even stop lingual tipping if the incisors are flared [32]. The compensation pattern of U1 is instantaneous, whereas the compensation in L1 shows a substantial effect in the first SD; however, no further compensation takes place in the second SD difference. Until now, no explanation has been reported.

From the multiple linear regression equation for U1 degree, the maxilla (SNA) and jaw discrepancy (ANB) revealed a negative relationship, which implies that a greater posterior position of the maxilla leads to a greater jaw discrepancy and an increase in the incisor compensation of U1. It was reported [33] that skeletal Class III with a declining ANB is likely to be accompanied by maxillary incisor proclination that is compatible with the features of malocclusion [17,22].

Only jaw discrepancy (ANB) showed a positive relationship with the L1 degree, which implies that greater jaw discrepancy results in a greater increase in the incisor compensation of L1. Unfortunately, no mandibular position parameter was presented in this equation. The findings suggest that the compensation of L1 cannot be explained by the mandibular location [34], which indicates that the incisor compensation of the mandibular incisors is closely related to the skeletal discrepancies in skeletal Class III patients.

The limitations of the study were the age of the subjects, the lack of subgroups of RMx+RMd and RMx(2SDs)+OMd, and the vertical skeletal patterns (FH-MP). The study did not include growing subjects because (1) the maxilla and mandible are still developing and cease to develop at different times and (2) the maxillomandibular relationship can change, which affects incisor compensation. A lack of subgroups in RMx+RMd and RMx(2SDs)+OMd meant no information was available for these subgroups. The vertical skeletal patterns (FH-MP) in Table 2 indicated specific divergence in each subgroup. Therefore, the results of incisor compensation cannot be applied to patients with different FH-MP.

This study contributes to diagnostics in terms of incisor compensation in specific subgroups, which was found in all subgroups except for U1 in PMx+PMd and L1 in OMx+OMd. Therefore, the diagnostic accuracy of incisor compensation in these two subgroups must be assessed with caution. Information concerning incisor compensation can be useful for treatment planning as to whether or not incisor decompensation should be applied in specific groups of orthognathic surgery patients. Also, the amount of incisor inclination adjustment should be considered in camouflage treatment for skeletal Class III patients based on the degree of incisor compensation. Class III camouflage patients usually present with a negative overjet; thus, orthodontists need to move U1 labially and L1 lingually to create a normal overjet. Therefore, either the PMx+PMd group with normal U1 inclination or the OMx+OMd group with L1 normal inclination could allow the clinicians to move either U1 labially or L1 lingually further than the incisors in the other groups that already have a proclined U1 or retroclined L1.

This study was a sagittal cephalometric investigation that did not include model measurement data, vertical cephalometric measurements, or soft tissue function factors that report the relationships relative to incisor inclination. Further research could focus on these factors: extent of crowding [35], level of dental malocclusion [36], morphology of the dental arch [37], vertical facial morphology [38], presence of open bite [39], occlusal plane angle [40], deviation [41], lip pressure [42], and tongue pressure [43]. Further studies for a better understanding of incisor compensation should include these factors. Furthermore, clinical studies regarding either decompensation in orthognathic surgery or camouflage orthodontic treatment to understand the changes in incisor decompensation after these treatments would be interesting. Moreover, further studies could be useful for esthetics since incisor inclination is related to the facial profile [44,45].

## 5. Conclusions

Maxilla (SNA) and jaw discrepancy (ANB) were inversely related to the U1 degree, whereas only jaw discrepancy (ANB) was positively related to the L1 degree. U1 in PMx+PMd and L1 in OMx+OMd showed no incisor compensation. U1 had limited compensation even with progressive maxillary retrognathism while L1 showed limited compensation after one SD mandibular prognathism.

## Figures and Tables

**Figure 1 diagnostics-14-01021-f001:**
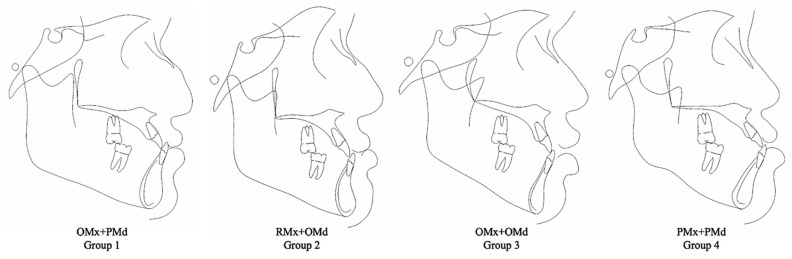
Illustrations of the skeletal and dental characteristics of each subgroup: Group 1, orthognathic maxilla and prognathic mandible (OMx+PMd); Group 2, retrognathic maxilla and orthognathic mandible (RMx+OMd); Group 3, orthognathic maxilla and mandible (OMx+OMd); and Group 4, prognathic maxilla and mandible (PMx+PMd).

**Table 1 diagnostics-14-01021-t001:** Definitions of the cephalometric variables used for analyzing the skeletal and dental aspects in this study.

Abbreviation	Definition
Skeletal	
SNA (degree)	This angle is formed by a line through the SN plane and point A, which represents the relative anteroposterior position of the maxilla to the cranial base.
SNB (degree)	This angle is formed by a line through the SN plane and point B, which represents the relative anteroposterior position of the mandible to the cranial base.
ANB (degree)	This angle is formed by lines NA and NB and represents the relative anteroposterior position of the maxilla to the mandible. This measurement is commonly used in orthodontics to determine skeletal classification.
FH plane	The Frankfort horizontal plane is defined as a line passing through the lowest point of the lower margin of the orbitale (Or) and the uppermost points of Porion (Po).
PP plane	The palatal plane is a horizontal plane that passes through the points of the posterior nasal spine (PNS) and the anterior nasal spine (ANS).
MP plane	The mandibular plane is a horizontal plane that passes through the points Menton (Me) and Gonion (Go).
FH-MP (degree)	This is the angle between the Frankfort Horizontal plane and the mandibular plane.
Dental	
U1-NA (degree)	This angle is between the long axis (incisal edge to the apex of the root) of the maxillary central incisors (U1) and the NA plane.
U1-NA (mm)	The linear distance between the tip of the upper central incisor (U1) perpendicular to the NA line in millimeters.
L1-NB (degree)	This is the angle between the long axis of the mandibular incisors and the NB plane.
L1-NB (mm)	The linear distance between the tip of the lower central incisor (L1) perpendicular to the NB line in millimeters.
U1-L1 (degree)	This angle is between the long axis of the maxillary and mandibular central incisors.
U1-PP (degree)	This angle is between the long axis of the maxillary central incisors and the palatal plane.
L1-MP (degree)	This angle is between the long axis of the mandibular central incisors and the mandibular plane.
OB (mm)	This is the vertical overlap of the incisors.
OJ (mm)	This is the horizontal overlap of the incisors.

**Table 2 diagnostics-14-01021-t002:** Baseline parameters among different subgroups of skeletal Class III.

Variables	Skeletal	OMx+PMd	RMx+OMd	OMx+OMd	PMx+PMd
	Class I (SI)	Group 1	Group 2	Group 3	Group 4
Frequency	30	34	21	62	20
Proportions	N/A	24.82%	15.32%	45.26%	14.60%
Age	24.86 ± 2.65	25.23 ± 4.62	24.25 ± 2.36	25.01 ± 4.98	23.38 ± 2.56
Male/Female	14/16	15/19	10/11	29/33	9/11
SNA	83.27 ± 3.17	83.88 ± 1.30	76.75 ± 2.38	82.68 ± 1.85	92.60 ± 2.40
SNB	81.85 ± 3.77	91.92 ± 1.98	80.92 ± 2.43	83.35 ± 1.14	92.20 ± 2.99
ANB	3.76 ± 1.14	−8.08 ± 1.98	−4.17 ± 2.86	−0.67 ± 2.11	0.40 ± 0.59
FH-MP	22.54 ± 3.61	23.00 ± 4.51	24.50 ± 4.48	27.53 ± 5.36	29.40 ± 5.32

OMx+PMd: orthognathic maxilla and prognathic mandible; RMx+OMd: retrognathic maxilla and orthognathic mandible; OMx+OMd: orthognathic maxilla and mandible; PMx+PMd: prognathic maxilla and mandible; SI: skeletal Class I.

**Table 3 diagnostics-14-01021-t003:** Cephalometric parameters among subgroups of skeletal Class III.

Variables	Skeletal Class I (SI)	OMx+PMd Group 1	RMx+OMd Group 2	OMx+OMd Group 3	PMx+PMd Group 4
U1-NA (degree)	23.25 ± 4.07	30.92 ± 4.26	35.92 ± 4.56	32.05 ± 5.48	28.0 ± 5.79
U1-NA (mm)	5.32 ± 1.23	7.46 ± 2.99	8.42 ± 3.73	7.53 ± 2.20	5.80 ± 1.25
L1-NB (degree)	32.03 ± 3.43	20.08 ± 5.11	21.08 ± 4.52	28.49 ± 6.52	24.80 ± 0.79
L1-NB (mm)	6.35 ± 1.32	5.04 ± 2.99	6.17 ± 4.17	7.40 ± 3.44	7.80 ± 0.62
U1-L1 (degree)	124.09 ± 4.75	134.00 ± 7.41	125.25 ± 5.71	121.33 ± 5.50	116.80 ± 3.23
U1-PP	119.46 ± 3.94	123.84 ± 7.11	123.25 ± 5.15	123.32 ± 4.88	123.80 ± 3.09
L1-MP	97.04 ± 3.79	82.88 ± 5.04	84.75 ± 5.07	88.75 ± 5.77	89.80 ± 2.25
OB	1.67 ± 0.67	2.44 ± 2.52	2.00 ± 2.00	0.75 ± 1.34	0.60 ± 0.40
OJ	1.66 ± 0.40	−2.60 ± 2.18	−2.08 ± 1.98	−1.32 ± 1.21	−2.00 ± 1.30

OMx+PMd: orthognathic maxilla and prognathic mandible; RMx+OMd: retrognathic maxilla and orthognathic mandible; OMx+OMd: orthognathic maxilla and mandible; PMx+PMd: prognathic maxilla and mandible; SI: skeletal Class I.

**Table 4 diagnostics-14-01021-t004:** Comparisons of *p*-values between groups.

Variables	SI-1	SI-2	SI-3	SI-4	1–2	1–3	1–4	2–3	2–4	3–4
U1-NA (degree)	<0.001 *	<0.001 *	<0.001 *	0.184	0.086	1.000	0.534	0.490	0.001 *	0.020 *
U1-NA (mm)	0.002 *	<0.001 *	<0.001 *	1.000	1.000	1.000	0.058	1.000	0.005 *	0.008 *
L1-NB (degree)	<0.001 *	<0.001 *	0.083	0.001 *	1.000	<0.001 *	0.159	0.004 *	1.000	0.325
L1-NB (mm)	1.000	1.000	1.000	<0.001 *	1.000	0.027 *	<0.001 *	1.000	0.014 *	0.007 *
U1-L1 (degree)	0.002 *	1.000	0.611	<0.001 *	0.223	<0.001 *	<0.001 *	0.682	0.002 *	0.024 *
U1-PP	0.019 *	0.602	0.002 *	0.019 *	1.000	1.000	1.000	1.000	1.000	1.000
L1-MP	<0.001 *	<0.001 *	<0.001 *	0.001 *	1.000	<0.001 *	0.006 *	0.436	0.480	1.000
OB	1.000	1.000	<0.001 *	0.025 *	1.000	<0.001 *	0.040 *	0.151	0.578	1.000
OJ	<0.001 *	<0.001 *	<0.001 *	<0.001 *	1.000	0.058	1.000	1.000	1.000	0.742

* Statistically significant difference (*p* < 0.05) by the Kruskal–Wallis test followed by Bonferroni tests. SI: skeletal Class I; Group 1: OMx+PMd; Group 2: RMx+OMd; Group 3: OMx+OMd; Group 4: PMx+PMd.

**Table 5 diagnostics-14-01021-t005:** Cephalometric parameters among different degrees of mandibular prognathism with orthognathic maxilla.

Variables	OMx+OMd	OMx+PMd (1 SD)	OMx+PMd (2 SDs)
Frequency	62	22	12
Proportions	64.59	22.91	12.50
SNA	82.68 ± 1.85	83.55 ± 1.60	83.83 ± 1.03
SNB	83.35 ± 1.14	87.55 ± 1.10	91.92 ± 1.93
ANB	−0.67 ± 2.11	−4.00 ± 1.57	−8.08 ± 1.98
FH-MP	27.53 ± 5.36	23.00 ± 5.51	23.00 ± 4.51
U1-NA (degree)	32.05 ± 5.48	30.95 ± 3.67	30.08 ± 4.56
U1-NA (mm)	7.53 ± 2.20	7.20 ± 2.21	7.50 ± 3.50
L1-NB (degree)	28.49 ± 6.52	20.64 ± 5.88	19.25 ± 4.05
L1-NB (mm)	7.40 ± 3.44	5.68 ± 2.61	4.92 ± 3.55
U1-PP	123.32 ± 4.88	123.73 ± 6.68	123.75 ± 5.83
L1-MP	88.75 ± 5.77	83.50 ± 6.36	81.58 ± 3.24

OMx+PMd: orthognathic maxilla and prognathic mandible; RMx+OMd: retrognathic maxilla and orthognathic mandible; OMx+OMd: orthognathic maxilla and mandible; PMx+PMd: prognathic maxilla and mandible; SD: standard deviation.

**Table 6 diagnostics-14-01021-t006:** Comparative *p*-values between the standard deviations of the prognathic mandible groups.

Variables	OMx+OMd vs. OMx+PMd (1 SD)	OMx+OMd vs. OMx+PMd (2 SDs)	OMx+PMd (1 SD) vs. OMx+PMd (2 SDs)
SNA	0.140	0.113	0.896
SNB	<0.001 *	<0.001 *	<0.001 *
ANB	<0.001 *	<0.001 *	<0.001 *
FH-MP	0.004 *	0.031 *	1.000
U1-NA (degree)	0.682	0.465	0.889
U1-NA (mm)	0.868	0.999	0.943
L1-NB (degree)	<0.001 *	<0.001 *	0.819
L1-NB (mm)	0.118	0.063	0.810
U1-PP	0.956	0.969	1.000
L1-MP	<0.001 *	<0.001 *	0.819

* Statistically significant difference (*p* < 0.05) according to the Kruskal–Wallis test followed by Bonferroni tests. OMx+OMd: orthognathic maxilla and mandible; OMx+PMd: orthognathic maxilla and prognathic mandible; SD: standard deviation.

## Data Availability

The data presented in this study are available upon request from the corresponding author.

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
