# Peer review of "Is Incisor Compensation Related to Skeletal Discrepancies in Skeletal Class III? A Retrospective Cephalometric Study"

_diagnostics, 2024, doi:10.3390/diagnostics14101021_

Round 1
Reviewer 1 Report
Comments and Suggestions for Authors
This paper examines the relationship between maxillo-mandibular morphology and the anterior tooth axis in Class III and is of great clinical interest and utility.
I would like to ask some questions to refine and make this paper more useful.
1、If you think that crowding amount affects the incination of maxillary anterior teeth in the maxilla arch, what do you think in this study?
2, This study is classified only in terms of horizontal relationship, but what do you think the results will be affected by vertical facial morphology as well?
Reviewer 2 Report
Comments and Suggestions for Authors
INTRODUCTION
1. First sentence: "Skeletal class III is a malocclusion characterized by maxillary and mandibular discrepancies". However, skeletal class II is also characterized by maxillary and mandibular discrepancies, so please evaluate more what kind of discrepancies are typical for skeletal class III. The sentence from lines 34-35 should be next to the first sentence.
2. Lines: 42-43 "Unfortunately..." - the aim of introduction is not a place for personal opinion of the authors, but to briefly present the current state of knowledge. Please rewrite.
3. Lines 53-60 include information that is presented earlier or that should be moved to discussion section. This paragraph should be rewritten.
4. Lines 60-69 are unclear. Line 62 - about what controversy are you writing? I see no controversy in the dental compensation of skeletal malocclusion. Line 64 - "grouped according to clinical practice" - what do you mean? This part must be rewritten or excluded from the text as it is very confusing.
5. The introduction should be finished with clearly stated aim of the study. This must be added.
MATERIALS AND METHODS
1. Were the patients with systemic illnesses included in or excluded from the study? How about endocrinopathies? If only generally healthy people were included into the study, this information must be added.
2. Lines 99-102 - if you hadn't done something, this shouldn't be described. This kind of information ought to be placed in the "limitation to the study" section.
3. Tithe to table 1 MUST be changed. Abbreviations, parameters, and definitions - but of what? this must be added.
4. The definition of ANB angle from table 1 is misleading. ANB angle is the angle between the lines NA and NB.
5. The definition of U1-NA (mm) from table 1 is unclear. What kind of distance? perpendicular to what line? shortest? longest? The same with L1-NB (mm) description.
6. You should carefully go through all of the definitions and verify the explanations given as many of them are not precise enough, i.e. when you use the word "axis" - which axis do you mean? In the scientific article everything must be clearly explained so that the readers have no doubts about the authors' methodology.
7. Paragraph 2.4 must be rewritten as it should only include general information about statistical analyses that the authors used. There should be no results in this section.
RESULTS
1. The statistical significance should be marked in the tables so that it is easily seen which values differ significantly.
DISCUSSION
1. Within the line 275 the authors pointed out only two limitations to the study. I recommend adding the "limitations to the study" section in which the authors think and discuss the limitations to the study and how they could have affected the final results.
2. I recommend adding to the discussion the role of sagittal position of lower incisors in facial profile esthetics:
Derwich M, Minch L, Mitus-Kenig M, Zoltowska A, Pawlowska E. Personalized Orthodontics: From the Sagittal Position of Lower Incisors to the Facial Profile Esthetics. J Pers Med. 2021 Jul 22;11(8):692. doi: 10.3390/jpm11080692.
Contini E., Orthod D., Campi S., Caprioglio A. Profile changes following lower incisor repositioning: A comparison between patients with different growth pattern. Minerva Stomatol. 2015;64:75–85.
COCNLUSIONS
This must not be repetition of the results, nor the methodology. This section should include major conclusions from the article. This part must be rewritten.
General comment: the manuscript requires major revision. The authors should focus on the above listed suggestions and comments.
Comments on the Quality of English LanguageExtensive editing of English language is mandatory.
Reviewer 3 Report
Comments and Suggestions for Authors
Re: diagnostics-2921369
Is incisor compensation related to skeletal discrepancy in skeletal class III?: A retrospective cephalometric study
This study examined the relationship between the position of the upper and lower jaw bones and the anterior teeth. The inclination of the anterior teeth must be related to various factors such as the level of dental malocclusion, the morphology of the dental arch, lip pressure, and the presence of open bite. It is also assumed to be affected by cases of deviation and occlusal plane angles. The relevance of these factors to the results of this study is unclear. It is also unclear how the results of this study will contribute to diagnosis and treatment selection for skeletal class III.
Reviewer 4 Report
Comments and Suggestions for Authors
This is a good-quality article with sufficient and concise literature review, sound methodology including objective sample size calculation, well-presented results, necessary and sufficient discussion along with a full list of key references. Only minor concern is that the article needs professional English editing.
Comments on the Quality of English LanguageOnly minor concern is that the article needs professional English editing.
Round 2
Reviewer 2 Report
Comments and Suggestions for Authors
The manuscript has been significantly improved.
Comments on the Quality of English LanguageExtensive English correction is needed.
Reviewer 3 Report
Comments and Suggestions for Authors
This is the first comment. And this was not resolved.
